# Molecular Markers to Predict Prognosis and Treatment Response in Uterine Cervical Cancer

**DOI:** 10.3390/cancers13225748

**Published:** 2021-11-17

**Authors:** Maximilian Fleischmann, Georgios Chatzikonstantinou, Emmanouil Fokas, Jörn Wichmann, Hans Christiansen, Klaus Strebhardt, Claus Rödel, Nikolaos Tselis, Franz Rödel

**Affiliations:** 1Department of Radiation Oncology, University Hospital Johann Wolfgang Goethe University, 60590 Frankfurt, Germany; Georgios.Chatzikonstantinou@kgu.de (G.C.); Emmanouil.Fokas@kgu.de (E.F.); ClausMichael.Roedel@kgu.de (C.R.); Nikolaos.Tselis@kgu.de (N.T.); Franz.Roedel@kgu.de (F.R.); 2German Cancer Research Center (DKFZ), 69120 Heidelberg, Germany; 3German Cancer Consortium (DKTK), Partner Site Frankfurt am Main, 60590 Frankfurt, Germany; strebhardt@em.uni-frankfurt.de; 4Frankfurt Cancer Institute, 60590 Frankfurt, Germany; 5Institute for Radiation Therapy and Special Oncology, Hannover Medical School, 30625 Hannover, Germany; Wichmann.Joern@mh-hannover.de (J.W.); Christiansen.Hans@mh-hannover.de (H.C.); 6Department of Gynecology and Obstetrics, University Hospital, 60590 Frankfurt, Germany

**Keywords:** cervical cancer, predictive, prognostic, molecular marker, biomarker, outcome, survival, response, chemoradiotherapy

## Abstract

**Simple Summary:**

Despite the implementation of efficient screening and vaccination programs, uterine cervical cancer remains a leading cause of cancer-related mortality in women worldwide. New therapeutic approaches have so far failed to improve treatment response and prognosis significantly, especially in patients with recurrent disease or metastases. Further, robust molecular markers to predict therapy response and survival are scarce and their routine use is limited in clinical practice. Accordingly, there is an urgent need to identify and establish molecular markers to predict therapy response and clinical outcome to improve treatment and survival in cervical cancer.

**Abstract:**

Uterine cervical cancer is one of the leading causes of cancer-related mortality in women worldwide. Each year, over half a million new cases are estimated, resulting in more than 300,000 deaths. While less-invasive, fertility-preserving surgical procedures can be offered to women in early stages, treatment for locally advanced disease may include radical hysterectomy, primary chemoradiotherapy (CRT) or a combination of these modalities. Concurrent platinum-based chemoradiotherapy regimens remain the first-line treatments for locally advanced cervical cancer. Despite achievements such as the introduction of angiogenesis inhibitors, and more recently immunotherapies, the overall survival of women with persistent, recurrent or metastatic disease has not been extended significantly in the last decades. Furthermore, a broad spectrum of molecular markers to predict therapy response and survival and to identify patients with high- and low-risk constellations is missing. Implementation of these markers, however, may help to further improve treatment and to develop new targeted therapies. This review aims to provide comprehensive insights into the complex mechanisms of cervical cancer pathogenesis within the context of molecular markers for predicting treatment response and prognosis.

## 1. Introduction

Transient infections with human papillomavirus (HPV) are common, with a lifetime probability ranging between 50% and 95% [1]. Persistent infections with oncogenic high-risk HPV16/18 subtypes frequently progress to cervical intraepithelial neoplasia (CIN), a precursor lesion to invasive uterine cervical cancer [2,3]. As a consequence of effective screening and vaccination programs, cervical cancer is largely preventable, although vaccination rates remain low and only a minority of women have access to efficient health care services [4]. Accordingly, more than half a million women worldwide are diagnosed with cervical cancer each year, with the majority of deaths occurring in low- and middle-income countries, indicating substantial global disparities [5,6]. The treatment of cervical cancer can be roughly divided into two central therapy options. For localized disease, without the detection of lymph node metastases or the presence of risk factors, surgery is the preferred treatment modality. The scope of these curatively intended interventions ranges from fertility preserving conization to radical hysterectomy or total mesometrial resection (TMMR), including extensive pelvic and paraaortic lymphadenectomy. In cases of locally advanced cervical cancer (Fédération Internationale de Gynécologie et d’Obstétrique (FIGO) ≥ IB3), definitive chemoradiotherapy (CRT) followed by brachytherapy (BT) is the standard of care [7]. BT is an integral part of definitive treatment regimens for patients with locally advanced cervical cancer [8]. It delivers a highly conformal dose distribution of radiation, maximizes local control and minimizes toxicities by adequately covering the remaining extent of disease and optimally sparing adjacent organs at risk [9,10,11]. Concurrent chemotherapy has significantly improved the survival of cervical cancer patients compared to radiotherapy alone [12,13]. Platinum-based chemotherapy regimens are, therefore, generally preferred for patients who are suitable for this as first-line treatment [14,15]. Further, a combination of cisplatin, paclitaxel or topotecan plus the VEGF (vascular endothelial growth factor) inhibitor bevacizumab showed a significantly improved the survival rate, but was also associated with increased toxicity [16,17]. More recently, immunotherapies have transformed the management of many solid tumors and is constantly evolving. In 2018, the American Food and Drug Administration (FDA) approved programmed death 1 (PD-1) inhibitor pembrolizumab for patients with recurrent or metastatic cervical cancer whose tumors express PD-L1 (combined positive score (CPS) ≥1) and with disease progression after chemotherapy [18]. Pending new evidence, numerous immune checkpoint inhibitors are currently being investigated in various settings for the treatment of cervical cancer, including the CALLA (durvalumab), NCT02635360 (pembrolizumab), and NRG-GY017 (atezolizumab) trials. Moreover, therapeutic vaccination strategies (e.g., NCT02853604) and adoptive cell therapies (e.g., NCT03108495) are currently being tested [19]. In this regard, the FDA recently approved the antibody-drug conjugate (ADC) tisotumab vedotin against tissue factor and monomethyl auristatin E (MMAE) for recurrent or metastatic cervical cancer after efficacy, and safety was demonstrated in the NCT03438396 multicenter, open-label, single arm, phase 2 trial (innovaTV 204/GOG-3023/ENGOT-cx6) [20]. Despite these advances, survival rates have not improved significantly over the past two decades and especially the prognosis for women with persistent, recurrent or metastatic disease remains poor. In addition, valid and easily accessible molecular markers for risk stratification and to predict treatment response and survival remain elusive. These markers, however, may cover a prerequisite to unravel the molecular pathogenesis and heterogeneity of cervical cancer and will be crucial to further develop novel therapeutic targets and treatment approaches.

## 2. Clinical Parameters to Predict Outcome and Response

Surgical procedures are the standard of care for women with early-stage disease and low-risk features [21]. For those patients, clinical characteristics and postoperative surgicopathologic risk factors were identified to predict local and distant control. Tumor size, stromal invasion, lymphovascular space involvement (LVSI), pathologically confirmed lymph node metastases, extension into parametrial tissue or positive surgical margins are considered to predict an intermediate or high risk of recurrence after primary surgery (Figure 1) [22,23]. According to these clearly interrelated characteristics, Sedlis’ respectively Peters’ criteria (Table 1) were developed to decide whether postoperative radiotherapy with or without systemic therapy is required to reduce the risk of recurrence [24,25]. For intermediate risk patients, adjuvant RT was associated with a 46% reduction of recurrence and a significantly lower risk of disease progression at 5 years, although the improvement in overall survival did not reach statistical significance [26,27]. Patients with more than three positive lymph nodes after surgery are more likely to have extra-pelvic recurrence than patients without lymph node involvement or less than three positive lymph nodes [28]. Therefore, cisplatin-based CRT following radical hysterectomy was associated with significantly improved progression-free survival (63% vs. 80%, HR 2.01, *p* = 0.003) and overall survival (71% vs. 81%, HR 1.96, *p* = 0.007) compared to RT alone, as demonstrated by the GOG 109 landmark trial in a cohort of 268 women with high-risk cervical cancer [24].

Importantly, prognostic factors evaluated for women with locally advanced cervical cancer treated with concurrent CRT were similar to those after primary surgery. Tumor size, FIGO stage, pelvic or paraaortic lymph node involvement, non-squamous-cell carcinoma, performance status and ethnical race were significantly correlated with survival [29,30,31,32,33]. Total treatment time should not be longer than eight weeks [34,35]. Moreover, the omission of brachytherapy has a stronger negative effect on survival than the exclusion of chemotherapy [36]. To achieve a maximum of pelvic control, the combined prescription EQD2 of external beam radiotherapy (EBRT) and brachytherapy should be greater than 80 Gy, while higher doses can be considered for tumors with poor response to EBRT or extensive disease [37,38,39]. Meanwhile, the use of intensity modulated radiotherapy (IMRT) generated lower rates of acute and late toxicity and favorable disease-specific survival [40,41,42].

### Hematological Parameters

Multiple hematological markers are reported to predict response to CRT as easily accessible and feasible biomarkers. In particular, lower pre-treatment and nadir hemoglobin levels were discussed to negatively impact on treatment response and survival [43]. In this regard, anemia is linked to tumor hypoxia and neo-angiogenesis, which is associated with an impaired local tumor control [44]. However, in a retrospective analysis of 2454 cervical cancer patients by Bishop et al., the impact of anemia was relativized. Anemia was not confirmed as an independent prognostic factor and only hemoglobin levels below 10 g/dL during RT were associated with an inferior disease-specific survival [45]. Until now, data about the predictive value of anemia remain inconclusive.

In addition, further cell lines of the blood count were evaluated. Cancer-related systemic inflammation is associated with impaired outcomes in patients with malignant diseases [46]. Tumor-induced neutrophilia and thrombocytosis, which may be considered a sign of systemic inflammation, and relative lymphocytopenia, which occurs as part of the antitumor immune response, are frequently observed [47]. In a meta-analysis, an elevated pretreatment neutrophil to lymphocyte ratio was confirmed as an independent predictor of a decreased OS and PFS, regardless of stage or primary treatment [48]. In addition, a systemic immune inflammation index based on peripheral neutrophil, lymphocyte and platelet counts was established in cervical cancer patients after primary surgery. An increased immune-inflammation index was significantly correlated with poorer OS regardless of FIGO stage [49].

Moreover, C-reactive acute phase protein (CRP), an inflammatory marker, and serum albumin levels indicative for liver function and malnutrition were associated with impaired immune response in cervix cancer patients [50]. The pretreatment CRP/albumin ratio was further identified as a predictor of survival in two studies. He et al. further concluded an increased predictive value from a combination of CRP/albumin ratio and neutrophil to lymphocyte ratio [51,52]. Low serum albumin levels were associated with a low nutritional index and predicted a poor prognosis in a cohort of 131 cervical cancer patients treated with RT or CRT [53]. The negative impact of a low nutritional index was further confirmed in a larger and more recent study [54]. Sarcopenia resulting from malnutrition, however, does not affect survival before initiation of treatment, whereas significant loss of skeletal muscle during CRT is associated with an impaired outcome [55,56,57].

Finally, elevated pretreatment levels of serum squamous cell carcinoma antigen (SCC Ag) were associated with a larger tumor size, the involvement of lymph nodes, LVSI and deep stromal invasion [58,59,60,61] and were confirmed to significantly correlate with an inferior survival [62].

## 3. Human Papillomavirus in Cervical Cancer

Cervical cancer is a prime example of a malignancy caused by infection with human papillomaviruses (HPV). Of at least 13 oncogenic high-risk subtypes from the same phylogenetic lineage, HPV16 and HPV18 are the predominant subtypes associated with disease onset and progression [63].

### 3.1. HPV-DNA Integration

The integration of HPV into the host genome causes a linearization of the viral genome and results in a partial or complete deletion of the viral genes E1 and E2. This further suggests that both integrated viral DNA and its episomal form can be present [64]. The status of the viral E2 gene was evaluated for its influence on therapy response. As a regulator of viral genome replication and RNA transcription, loss of E2 function after viral integration leads to a more radioresistant phenotype by potentiating E6 and E7 expression and reducing pro-apoptotic signaling [65]. In line with this, there is evidence suggesting a favorable prognosis in patients with HPV 16+ tumors and an intact E2 gene [66,67], while episomal expression of E2, E4 and E5 results in enhanced proliferation and increased susceptibility to induction of cancer [68]. In a prospective cohort of 272 HPV+ cervical cancer patients enrolled in the BioRAIDs study, several integration signatures of the HPV genome were identified. The most frequent integration site was the MACROD2 (mono-ADP Ribosylhydrolase 2) gene locus, followed by TP63 and MIPOL1/TTC6 loci. Although there is a lack of evidence for the impact of the integration within MACROD2, MACROD2 deletions are associated with impaired PARP1 (poly(ADP-ribose)polymerase 1) activity and chromosomal instability in other entities. Notably, the integration of HPV DNA was much more frequent in cervical cancer compared with anal cancer and was associated with a low HPV copy number (<4, ratio of the number of HPV reads over the control human kallikrein-*3* (KLK3) gene) and worse progression-free survival. Episomal forms of HPV DNA were associated with a high HPV number and phosphatidylinositol-4,5-bisphosphat-3-kinase (PIK3) mutations [69].

### 3.2. Prognostic Impact of HPV Viral Load

Determining the predictive value of the HPV viral load was the objective of several studies. In locally advanced cervical cancer, a low HPV viral load was associated with an impaired distant metastasis free (DMFS) and overall survival (OS), while the combined analysis HPV viral load and tumor size were superior for predicting OS [70]. These findings are consistent with the findings by Kim et al., who reported worse disease-free survival (DFS) in 169 women with low HPV viral load treated with CRT [71]. A low and persistent HPV viral load was further correlated significantly with poor local recurrence-free survival (LRFS) in another study covering 156 cervical cancer patients treated with CRT or radiotherapy (RT) alone [72]. Furthermore, in a cohort of 520 woman treated with surgery or CRT, a low HPV viral load predicted poor prognosis irrespective of treatment [73]. In contrast, at the tissue and cellular levels, Cao et al. hypothesized that a high HPV viral load influences the tumor microenvironment (TME) towards a more immunosuppressive milieu with an increased proportion of FOXP3-positive tumor-infiltrating regulatory T-lymphocytes (TILs), contributing to an impaired survival rate [74].

### 3.3. Genetic Alterations, Immune Responses and Angiogenesis in HPV+ Cervical Cancer

Nevertheless, a persistent HPV infection is not sufficient to induce carcinogenesis per se. Further genetic and epigenetic alterations and an interplay between infected cells, the host immune response and the TME are required to promote malignant transformation and propagation [75,76]. A recent genome-wide association study provided new evidence for a genetic disposition and susceptibility of cervical cancer. Bowden et al. identified six variants in the PAX8 (paired-box-protein 8), CLPTM1L and HLA (human leukocyte antigen) genes that were associated with increased risk of CIN and cervical cancer. These variants are associated with disruptions in multiple pro-apoptotic and immune function pathways [77] and are consistent with another genome-wide association study, which identified multiple loci affecting immune pathways of antigen presentation and immune checkpoints associated with increased risk of cervical cancer development [78]. Furthermore, variants of major histocompatibility complex (MHC) alleles are associated with increased risk of persistent HPV infection and cervical cancer [79].

The innate and adaptive immune response is crucial in clearing HPV infections and in surveillance of carcinogenesis and tumor progression [80,81]. Accordingly, there are several immune evasion mechanisms and immunomodulatory effects exploited by HPV oncoproteins that disrupt pivotal anticancer immune response pathways. In vitro and in vivo analyses of oncoproteins E6 and E7 revealed Toll-like receptor 9 (TLR9) downregulation and suppression of C-X-C motif chemokine ligand CXCL14, which in turn reduced the local immune response and the recruitment of antigen-presenting cells, natural killer (NK) and T cells [82,83]. A local immune response is further compromised by E7 targeting the stimulator of interferon gene (STING) pathway, which senses against intracellular DNA and activates the production of pro-inflammatory cytokines, such as interferon gamma (IFNγ) [84,85]. E6 and E7 interact with multiple IFN-receptor pathways and impede the phosphorylation of signal transducer and activators of transcription 1 (STAT1), STAT2 and other JAK/STAT pathways, which are critical for IFN signaling as well as proliferation and invasion [86]. Furthermore, activation and recruitment of immune components such as Langerhans cells, dendritic cells, tumor-associated macrophages, CD4+ and CD8+ lymphocytes are enhanced by HPV fostering of an immunosuppressive tumor microenvironment (TME) by a plethora of signaling pathways, including nuclear factor kappa B (NF-κB) signaling, antigen presentation and the aforementioned IFN signaling pathways [81].

Finally, HPV infection is linked to angiogenesis. In a TP53-dependent manner, the expression of thrombospondin 1, maspin and VEGF is altered by HPV E6 and E7 oncoproteins, enhancing angiogenesis [87]. Concerning TP53 independent pathways, Wang et al. showed an upregulation of ribonucleotide reductase small subunit M2 (RRM2) by E7 and enhanced angiogenesis mediated by the ROS-ERK1/2-HIF1α-VEGF axis [88].

### 3.4. Oncoproteins E6 and E7 Increase Radiosensitivity of HPV-Positive Cervical Cancer Cells

Importantly, HPV-associated malignancies are linked to an intrinsic radiation sensitivity, resulting in a favorable response to radiotherapy [89,90,91,92]. As summarized in Figure 2, following the integration of the HPV genome, expression of viral oncoproteins E6 and E7 impact on pivotal cellular pathways by inactivating the tumor suppressor protein TP53 and retinoblastoma (Rb) protein, respectively. In brief, E6-mediated inactivation of TP53 requires the ubiquitin ligase E6AP, subsequently influencing cell cycle, apoptosis, cellular stress response and genomic stability [93]. The degradation of pRb by E7 activates the transcription factor E2F1, which promotes the upregulation of several S-phase genes, including the cyclin-dependent kinase inhibitor p16^ink4a^. In turn, p16^ink4a^ is directly involved in radiation response by interfering with mechanisms of DNA damage response [94,95]. For instance, p16^ink4a^ suppresses nuclear RAD51 foci, a marker of homologous recombination functionality, via downregulation of cyclin D1 [96]. Moreover, a PCR (polymerase chain reaction) array of DNA damage response genes revealed significantly lower RAD51 and breast cancer type 1 susceptibility protein (BRCA1) expression, indicating an impaired DNA damage response repair capacity [97]. Targeting poly(ADP-ribose) polymerase (PARP) is an effective treatment option in treating BRCA1 or BRCA2 negative cancers [98]. PARP1 is thereby involved in non-homologous end-joining (NHEJ) pathway and single-strand DNA repair. Ijff et al. recently showed that PARP1 inhibition sensitizes cervical cancer cell lines for CRT [99]. In addition, DNA single-strand break repair is directly impaired by E6 binding X-ray repair cross-complementing protein 1 (XRCC1), while double-strand break repair fidelity is compromised in both aTP53-dependent and -independent manner [100,101]. Notably, HPV 18+ tumors are considered to be more aggressive and display an impaired prognosis following radiotherapy [102,103,104]. As a potential mechanism, expression patterns of four DNA repair genes in HPV 18+ tumors were identified. TP52BP1, MCM9 (minichromosome maintenance 9 homologous recombination repair factor), POLR2F and SIRT6 were associated with higher activity of NHEJ and homologous recombination (HR) pathway and increased DNA repair capacity [105].

## 4. Molecular Protein Markers

The Cancer Genome Atlas (TCGA) database has expanded the view of the genomic landscape of cancer by characterizing molecular patterns of many malignancies. Based on this database, subgroups of keratin-low and keratin-high squamous cell carcinomas and adenocarcinoma-rich, endometrial-like cervical cancer were identified. The endometrial-like subgroup, for instance, is associated with a high frequency of KRAS, ARID1A (AT-rich interaction domain 1A) and phosphatase and tensin homolog (PTEN) mutations and is predominantly HPV-negative. PIK3CA (phosphatidylinositol-4,5-bisphosphat-3-kinase, catalytic subunit alpha), PTEN and MPK1 (mitogen-activated protein kinase 1) were confirmed as significantly mutated genes (SMG), while ERBB3, CASP8, HLA-A and TGFBR were identified as novel SMGs in cervical cancer. Amplifications in programmed death 1 (PD-L1) and PD-L2 and a relatively high number of gene fusions in BCAR4 are providing possible targets for novel therapeutic approaches [106].

Genomic alterations in phosphoinositide 3-kinase (PI3K) are common in many different entities, including cervical cancer [107,108]. Activating mutations of PI3K and its catalytic subunit alpha (PIK3CA) are associated with co-alterations of PTEN, MAPK and AKT1, which are sufficient to foster tumorigenesis, tumor growth, migration, protein synthesis and glucose metabolism in pre-clinical models [109,110,111,112,113]. Tumor progression via downstream activation of the PI3K/AKT/mammalian target of rapamycin (mTOR) signaling cascade is thereby induced and further promoted by viral oncoproteins E5, E6 and E7 in HPV-associated malignancies [114]. Arjumand et al. indicated that cervical cancer cells with PIK3CA-E545K mutation are more resistant to cisplatin or cisplatin-based CRT than cells with PIK3CA wild type [115]. Although these aspects seem to influence the treatment response and survival of cervical cancer patients, the predictive value of PIK3CA mutations remains inconclusive. In a cohort of 89 patients with high-risk early-stage cervical cancer who underwent post-operative radiotherapy, a PIK3CA mutation was the most frequently detected aberration (29.2%) but displayed no prognostic impact [116]. In contrast, another study detected PIK3CA mutations in 37.3% of 161 pre-treatment specimens from cervical cancer patients treated with CRT, which was associated with worse OS in univariate analysis (*p* = 0.037) [117]. These discordant findings are consistent with a systemic review by Pergialiotis and colleagues that included twelve research articles and a total of 2196 cervical cancer patients and demonstrated no impact of PIK3CA mutations on treatment response and survival [118].

In a comprehensive review, Noordhuis et al. described in 2011 potential markers to predict response and survival of cervical cancer patients treated with CRT. Based on 42 eligible studies the authors identified 27 out of 82 markers that were independently associated with survival [119]. Some of these markers have been investigated in detail in other studies and will also be mentioned in the following paragraphs.

A hypoxic TME and a dysregulated neo-angiogenesis is a common feature in solid tumors. Activation of hypoxia-inducible factor 1 α (HIF-1α) upregulates VEGF, which in turn promotes angiogenesis, although also modulates TP53 activity and protects cells from hypoxia-induced apoptosis [120]. In cervical cancer, a high expression level of HIF-1α and its downstream target, the tumor-associated cell-surface glycoprotein carbonic anhydrase 9 (CA9), is linked with poor prognosis and therapy resistance [121,122,123]. In addition, overexpression of VEGF was correlated with a significantly worse OS and DFS/PFS with pooled hazard ratios of 2.29 and 2.77, respectively, in a meta-analysis including 1306 patients performed by Zhang et al. [124]. Notably, assessment of VEGF expression was performed with a variety of methods, meaning the overall results must be evaluated in a limited way considering the heterogeneity of the studies included. Additionally, women treated with (chemo)radiotherapy were under-represented.

As further described by Noordhuis et al., cyclooxygenase-2 (COX-2) and the epidermal growth factor receptor (EGFR) pathway could be associated with therapy response and outcome in patients with cervical cancer treated with (C)RT. COX-2 is one of two isoenzymes that catalyze the synthesis of prostaglandins from arachidonic acid and drive inflammation [125]. In cancer, COX-2 is frequently overexpressed and associated with angiogenesis, disease progression, metastatic behavior and therapy resistance [126,127,128]. Several groups have reported a negative prognostic and predictive impact of COX-2 expression in cervical cancer patients treated with (C)RT, while others did not [129,130]. In a recent retrospective analysis, COX-2 expression was significantly associated with LVSI but not with any survival endpoint [131]. Data from the RTOG 0128 trial, a phase I-II study testing the COX-2 inhibitor celecoxib and CRT in patients with locally advanced cervical cancer, revealed that patients with low COX-2 expression treated with celecoxib plus CRT had a worse OS compared to patients with a high COX-2 expression [132]. Nevertheless, this approach was not pursued further due to excessively high rates of acute toxicities [133].

As upstream modulators of COX-2, members of the mitogen-activated protein kinase (MAPK), extracellular-signal regulated kinase (ERK) family and EGFR may be indicators of outcome and survival in cervical cancer. While data on the impact of the MAPK pathway are limited, numerous studies exist on EGFR signaling in cervical cancer. The EGFR receptor transmits growth signals not only via the RAS-RAF pathway, but also via PIK3 and is linked to multiple signaling cascades [134]. EGFR protein expression and gene amplifications have been correlated with unfavorable clinical outcomes and poor response to (C)RT in cervical cancer [135,136,137,138]. A meta-analysis on the prognostic impact of EGFR overexpression pooled 22 studies and demonstrated that elevated levels of EGFR were associated with inferior OS and DFS. Moreover, EGFR overexpression was correlated with a higher incidence of lymph node metastases and tumor size [139]. Nevertheless, it is still controversial whether the degree of EGFR overexpression or gene amplification should be used as a reliable and robust marker [140]. At present, the use of EGFR inhibitors has been investigated in phase II studies in recurrent or metastatic cervical cancer patients and has shown minimal or no benefit at all [141,142,143,144].

Other proteins orchestrating EGFR signaling, including ERK, pAKT and PTEN, were not related to survival in patients with cervical cancer [133]. Instead, overexpression of CD24, an upstream of glycosylphosphatidylinositol-anchored protein (GPI) stimulating Akt, ERK, and nuclear factor kappa B (NF-κB), was associated with inferior OS in a small cohort of cervical cancer patients [145]. Additionally, the predictive and prognostic impact of KRAS mutations was evaluated in a cohort of 876 early-stage cervical cancer patients (FIGO IB1-IIA2). KRAS mutations were predominant in non-squamous cell carcinomas (8.2% vs. 2.2%, *p* < 0.001), associated with HPV18+ tumors (*p* = 0.003) and significantly correlated with a worse RFS in univariate and multivariate analyses. Notably, these results were not reproducible in squamous cell carcinomas and HPV16+ tumors [146].

Polo-like kinases (PLK) represent five highly conserved serine or threonine protein kinases participating in cell cycle regulation, DNA replication, mitosis and response to various types of cellular stress [147,148,149]. PLK1, the prototype member, is essential during mitosis and closely related to cellular proliferation and tumor growth. In cervical cancer, PLK1 overexpression was detected using RNA sequence datasets extracted from 290 cervical cancer tissue specimens. A high level of PLK1 expression had negative impact on survival, consistent with previous findings [150,151]. Yang et al. further demonstrated that phosphorylation of PLK1 by c-ABL regulates its activity, while knockdown of c-ABL inhibits downstream activation of Aurora A, a key kinase in mitotic progression. Again, c-ABL mediated phosphorylation of PLK1 was associated with a worse 5-year survival rate, while PLK1 mutant Y425 partially inhibits tumor growth in mice [152]. Upon ligand stimulation of CD95 (Fas receptor), PLK3 phosphorylates pro-caspase-8 on residue threonine 273 and promotes apoptosis in vitro [153]. Against this background, the prognostic impact of both PLK3 and pT273 Caspase-8 was investigated in patients with cervical cancer. High levels of PLK3 and pT273 Caspase-8 were associated with improved survival rates after definitive CRT [154]. In general, PLK3 is associated with cellular stress response, linking DNA damage to DNA damage repair, cell cycle arrest, apoptosis and cellular adhesion [148,149,155] (Figure 3).

The impacts of DNA damage response (DDR) proteins were investigated by a Canadian group. In a set of pre-treatment samples from 117 cervical cancer patients treated with CRT, the proteins ATM, PARP-1, DNA-PKcs, Ku70 and Ku86 were quantified using fluorescence immunohistochemistry. Low expression of ATM and PARP-1 was significantly associated with a worse 5-year PFS. Moreover, expression of ATM, PARP-1, DNA-PKcs and Ku86 was associated with a shorter OS, while multivariate analysis confirmed ATM and PARP-1 as independent predictors for PFS and OS. Accordingly, the authors hypothesized that the assessment of ATM status may also be a predictive biomarker of PARP-1 inhibitor treatment efficacy [156].

In conclusion, protein biomarkers reflect on the profound and complex genomic and cellular alterations of cervical cancer, although single markers are less likely to predict prognosis and treatment response at present. Small sample sizes, lack of reproducibility and the heterogeneity of biochemical methods further complicate their practical applicability in clinical practice. Combined marker analysis, however, may cover a possible approach to predict treatment response and survival more precisely. Consequently, Choi et al. assessed the expression levels of 22 proteins from pre-treatment biopsies of 181 locally advanced cervical cancer specimens and identified a panel of BCL2, HER2, CD133, CA9 and ERCC1 as an independent predictor of survival after CRT [157].

## 5. MicroRNA, Long-Non-Coding RNA and Circular RNA

MicroRNA (miRNA), long-non-coding RNA (lncRNA) and circular RNA (circRNA) encompass a variety of RNA molecules that regulate gene expression and a multitude of cellular signaling pathways. RNA molecules have emerged within a growing body of evidence as potential dynamic biomarkers to predict therapeutic response and survival, and may be useful for monitoring and detecting persistent or recurrent disease [158,159,160,161].

### 5.1. MicroRNAs

MicroRNAs can be classified into oncogenic and tumor suppressive miRNAs, which affect a variety of specific target genes and impact on cell growth, malignant transformation, cell migration, invasion and therapy resistance. Target genes are regulated by binding specific sequence motifs within the 3’ untranslated region (UTR) of mRNAs [162,163,164]. In cervical cancer, there are several types of endogenous and viral forms of RNA derived from the HPV genome that participate in carcinogenesis and tumor progression [165]. To identify consistent signatures of miRNA profiles, He at al. performed a comprehensive analysis of 85 studies including 2099 non-cancerous tissue samples, 827 CIN samples and 3095 samples from cervical cancer tissues. First, they identified 42 upregulated and 21 downregulated miRNAs from CIN to cervical cancer. In the following meta-analysis, five upregulated miRNAs (hsa-miR-10a-5p, −16-5p, −25-5p, −92a-3p and −196a-5p) and seven downregulated miRNAs (hsa-miR-29a, −34a, −99a-5p, −100-5p, 199a-3p, −203 and −218-5p) were confirmed to be dysregulated in CIN and cervical cancer. Inverse miRNA and mRNA expression levels revealed that the identified miRNA signature is involved in numerous cancer-associated pathways, such as MAPK-, WNT-, TGF-β- and p53-pathways, as well as cytokine receptor and extracellular matrix receptor pathways, regulating cell cycle, proliferation and apoptosis [166]. In another systematic review of 24 studies, some overlap (miR-29a and miR-21 were reported to be most frequently down and upregulated miRNAs in cervical cancer) with the analysis of He et al. was reported, although in conclusion different miRNAs were used to create a diagnostic and prognostic panel, indicating difficulties in validating a single miRNA signature [165]. Nevertheless, overexpression of miR-21, miR-34a, miR-196a, miR-27a and miR-221 was identified as a unique and specific miRNA signature in HPV+ squamous cell cervical cancer [167].

HPV oncoproteins E6 and E7 thereby influence miRNA expression itself. Via the degradation of TP53 by E6, HPV downregulates the tumor suppressor miRNA miR-34a [168,169]. The suppression of miR-424 by E6 and E7 further results in CHK1 induction targeting DNA damage repair [170], while low miR-424 expression is inversely correlated with CHK1 and p-CHK1 levels and associated with advanced FIGO stage, poor tumor differentiation, lymph node metastasis and LVSI [171]. Elevated levels of miR-31 were further identified as an independent prognostic factor associated with advanced FIGO stage, lymph node metastases and LVSI. In this study, Wang et al. further identified ARID1A as a direct target of miR-31 [172]. Notably, ARID1A loss is associated with trastuzumab resistance in breast cancer [173]. In addition, upregulation of miR-20a/b was correlated with increased proliferation, tumor growth, migration and invasion via AKT/p38 pathway activation and inhibition of tissue inhibitor of metalloproteinase 2 (TIMP-2), fostering epithelial–mesenchymal transition (EMT) [174,175]. Finally, overexpression of miR-375 increases TP53, TP21, Survivin and Bax expression in line with caspase-3 and -9 activity in HPV18+ cervical cancer cells. The suppression of the insulin-like growth factor-1 receptor (IGF-1R) could be another tumor suppressive mechanism exerted by miR-375 expression [176].

### 5.2. Long-Non-Coding RNAs

LncRNAs are encoded by a vast proportion of the human genome covering regulatory transcripts exceeding 200 nucleotides that are associated with disease and cancer development and progression by interacting with proteins, mRNAs and miRNAs [177,178,179]. Gene expression is regulated by cis-acting lncRNA or trans-acting lncRNAs, respectively. Cis-acting lncRNAs recruit regulatory factors to specific gene loci, influence transcription or splicing of genes or regulate adjacent genes and chromatin, while trans-acting lncRNAs regulate distant gene expression via the interaction with promoters, RNA-polymerases and other proteins [180].

HPV directly affects lncRNAs such as HOTAIR (HOX transcript antisense intergenic RNA) and MALAT1. HOTAIR is an lncRNA encoded by the antisense strand of the HOXC gene and controlled by HPV 16 E7. By recruiting the chromatin-remodeling complex PRC2, HOTAIR is regulating the expression of cancer-related pathway genes [181]. Moreover, HOTAIR sponges miRNAs such as miR-23b and miR143-3p, thereby modulating MAPK1 expression and the BCL2 axis, facilitating tumor cell proliferation and favoring metastatic behavior [182]. In several cervical cancer cell lines, HOTAIR overexpression was shown to have regulatory influence on additional signaling cascades such as the mTOR and the Notch-Wnt pathway [183,184]. Additionally, HOTAIR fosters the malignant potential via upregulation of VEGF, MMP-9 and other EMT-related genes. Therefore, HOTAIR overexpression is associated with the presence of lymph node metastases and inferior overall survival [185]. Remarkably, higher HOTAIR serum levels were also associated with advanced T-stage, lymph node metastasis and LVSI, thereby impairing survival [184].

The regulation of EMT-related genes is accomplished by lncRNAs such as metastasis-associated lung adenocarcinoma transcript 1 (MALAT1) [186]. The latter is overexpressed in cervical cancer cell lines and in HPV-associated cervical cancer tissues sponging several miRNAs [187]. Based on samples from 50 high-risk HPV+ cervical cancer patients, Lu et al. demonstrated that MALAT1 expression is significantly elevated in radiation-resistant tumors. Following radiation exposure, an inversely changed expression of MALAT1 and the tumor suppressive miRNA miR-145 further suggests their interaction and substantiates the hypothesis of the regulatory effect of miR-145 on cell cycle regulators CDK6, CDK2 and Cyclin D1 [188]. MALAT1 overexpression is further correlated with advanced tumors size and FIGO stage, LVSI, lymph node metastases and impaired OS [189].

The lncRNA cervical carcinoma high-expressed 1 (CCHE1) is seen as a regulator of the ERK/MAPK pathway and is considered a negative prognostic factor in cervical cancer [190,191]. Correlated with advanced FIGO stages and other clinicopathological risk factors, colon cancer-associated transcript 2 (CCAT2) is linked to a worse survival rate [192], while growth arrest-specific transcript 5 (GAS5) has been identified as a tumor-suppressive lncRNA in several entities including cervical cancer. On a mechanistic level, GAS5 may increase radiosensitivity in cervical cancer cells via miR-106b and upregulation of forkhead box O1 (FOXO1) and PTEN by sponging miR-196a and miR-205 [193,194,195].

Finally, Chen et al. established a set of six immune-related lncRNAs (AC009065.8, LINC01871, MIR210HG, GEMIN7-AS1, GAS5-AS1 and DLEU1) to define a risk score, which covers an independent prognostic signature in cervical cancer patients. Associated with the Wnt and TGFβ pathways, the group hypothesized that the lncRNA prognostic signature could help to identify subgroups of patients, which may benefit from immunotherapeutic interventions [196]. Despite all limitations, this work, therefore, represents an interesting approach towards personalized medicine.

### 5.3. Circular RNAs

CircRNAs are highly conserved, covalently closed RNA species that act in a regulatory network with miRNA and other inhibitory proteins, regulating and influencing gene expression, cancer progression, EMT and the tumor microenvironment (TME) [197]. Characteristic expression patterns in different types of cancer suggest circRNAs as tool to predict and monitor disease-specific survival endpoints [198,199]. Indeed, a variety of circRNAs are overexpressed and can function as oncogenes or tumor suppressors in cervical cancer [200]. In a microarray analysis of 35 cervical cancer tissues, 45 circRNAs were shown to be significantly dysregulated in cervical cancer. Most prominent hsa_circ_0018289 was upregulated in the cervical cancer tissues analyzed. In vitro and in vivo knockdown of hsa_circ_0018289 lead to decreased proliferation, migration and invasion of cervical cancer cells via sponging miR-497 [201]. Moreover, hsa_circ_0023404 overexpression was associated with a poor prognosis in cervical cancer and its knockdown was shown to suppress proliferation, cell cycle progression and migration in a report by Zhang et al. [202]. Other reports classified hsa_circ_0000263 as oncogenic circRNA affecting p53 in a regulatory network in cervical cancer, while hsa_circ_0001445 may act as a tumor suppressor due to its ability to sponge miR-620 [203,204]. The sequestration of miR-136 by hsa_circ_0023404 further promotes the yes-associated protein 1 (YAP) signaling pathway via transcription factor CP2 (TFCP2) and cancer progression, resulting in an impaired prognosis [202]. Hsa_circ_0023404 has also been associated with VEGF induction in cervical cancer tissues, which is related to increased metastasis rates and resistance to therapy [205]. Furthermore, a positive correlation with TNM stage, tumor size and lymph node metastases and a negative prognostic impact on survival is reported for hsa_circRNA_101996 expression [206]. Similar results are reported for hsa_ circ_0067934 sequesting miR-545 in a regulatory network affecting the EGFR axis [207]. Finally, in addition to endogenous circRNA, high-risk HPVs encode oncogenic viral circRNA. CircE7 was shown to translate for E7 oncoprotein in a CaSki cell line. Interestingly, knockdown of circE7 reduces tumor growth and malignant potential in vitro and in vivo [208].

## 6. Circulating Tumor Cells (CTC), Circulating Cell-Free DNA (cfDNA) and miRNA in Cervical Cancer

Advances in biotechnology techniques have enabled the detection of small amounts of RNA and DNA as well as tumor cells in a patient’s peripheral blood. Circulating molecules and markers represent a promising diagnostic, prognostic and dynamic tool and can be easily performed as a liquid biopsy [209,210].

### 6.1. Circulating Tumor Cells (CTCs)

The non-invasive, real-time collection of CTCs provides information about disease status and progression and may help identify potential biomarkers for predicting outcome and survival [211]. Analyzing blood samples (3.2 mL) from a cohort of 107 cervical cancer patients, Du et al. detected one to 27 CTCs in 86 patients. Those patients had a significantly shortened PFS. The negative impact of CTC detection on PFS was next confirmed in the multivariate analysis [212]. In another study, a combination of CTC count and serum SCC Ag levels was significantly associated with DFS in a cohort of 99 patients with locally advanced cervical cancer treated with (C)RT and remained significant in multivariate analyses [213]. Within the translational research of the randomized and prospective GOG 240 trial, the impact of the CTC count was evaluated in 176 patients with recurrent or metastatic cervical cancer. The findings indicated that a more pronounced decrease in the CTC count to be associated with a lower risk of death, while patients with a CTC count ≥ median revealed a significant survival benefit due to the addition of bevacizumab. The authors consequently speculated that a high (≥median) pretreatment CTC count might reflect increased neovascularization and vulnerability to VEGF inhibition [214].

Nevertheless, the impact of CTCs on cervical cancer remains inconclusive. This is primarily due to the various isolation methods used for the detection of CTCs, making reproducibility considerably more difficult, while large series for the validation of individual methods are not available at present.

### 6.2. Circulating Cell-Free Tumor DNA (ctDNA)

The isolation of circulating tumor DNA (ctDNA) from the patient’s peripheral blood allows a molecular characterization of the tumor and the identification of potential genetic alterations [215]. In general, higher ctDNAs levels are associated with advanced FIGO stage, tumor size, grading and lymph node metastases in small cohorts of patients with cervical cancer [210,216]. Comparable to an extensive molecular characterization of cervical cancer tissue, ctDNA provides information regarding mutational burden, amplifications and alterations. Several groups have identified common target gene mutations in PIK3CA, ALK, EGFR, ATM, BRCA2, ERBB2, APC, KRAS, BRAF, ABL1 and PTEN based on genomic analyses of ctDNA [217,218,219,220]. The prognostic relevance of these signatures was further evaluated by matching them with clinical data. For instance, Tian et al. identified a broad spectrum of mutated genes in a cohort of metastatic cervical cancer patients. The presence of any of these alterations was associated with an impaired PFS and OS. Patients with more than two metastases had a significantly higher mutational burden than patients with less than two metastases. In a serial ctDNA analysis, a decrease in the detected genetic alterations was associated with treatment response, while disease progression was inversely associated with a higher mutational burden [217].

### 6.3. Circulating HPV DNA

In a small cohort of 19 patients with metastatic cervical cancer, Kang et al. were able to detect HPV cfDNA in all samples, while other authors reported the detection of HPV related DNA from 6.9% up to 30% of invasive cervical cancer cases [221,222]. In sequential samples from patients with complete response to CRT, a complete clearance of HPV cfDNA became evident, while baseline HPV cfDNA levels had no impact on treatment response [223]. This assumption was further corroborated by a meta-analysis by Gu et al., who investigated the prognostic impact of HPV cfDNA in 684 patients from ten studies. All studies included had a high specificity in common, whereas sensitivity increased only in recent years. In summary, the meta-analysis revealed that HPV cfDNA levels in liquid biopsies are suitable for diagnostic and monitoring purposes. The relatively small sample size and the heterogeneity of the sampling methods, however, remains a problem in assessing the prognostic value of HPV cfDNA serum levels [224].

### 6.4. Circulating miRNAs

Several groups investigated singular circulating miRNAs or miRNA profiles in liquid biopsies. Low levels of miR-218 in the serum of 90 cervical cancer patients were associated with a higher incidence of lymph node metastases, a higher tumor volume and adenocarcinomas [225]. High levels of miR-205 were associated with large and less differentiated tumors and lymph node metastases resulting in worse OS [226,227]. Similar results were shown in small cohorts for serum levels of miR-20a and miR-101 [228,229]. Moreover, miR-21 detection in 89 cervical cancer patients was also associated with a significantly higher incidence of lymph node metastases. Qui et al. confirmed a significant correlation of miR-21 levels and lymph nodes metastases in a more recent study with 112 early-stage cervical cancer patients and 45 women diagnosed with CIN. Furthermore, high serum levels of miR-21 were associated with impaired RFS, while similar findings were described for low serum levels of miR-125b and miR-370 [230]. Jia et al. collected blood samples from 213 women with early-stage disease and identified five altered circulating miRNAs (miR-21, miR-29a, miR25, miR200a and miR486-5p) with miR-29a and miR-200a levels to correlate with advanced tumor stages and poor differentiation [231]. Interestingly, these results were consistent with the results from other groups investigating tissue samples from cervical cancer patients for miRNA expression [232].

## 7. Tumor Microenvironment in Cervical Cancer

A network of immune and endothelial cells, fibroblasts, signaling proteins and extracellular matrix molecules within the tumor and the surrounding tissue constitutes the TME [233]. In that context, immunosuppressive and immunogenic tumor-infiltrating immune cells influence the prognostic landscape in cervical cancer, which is highly related to and modulated by HPV [234] (Figure 4).

Tumor-infiltrating lymphocytes (TILs) interfering with immune checkpoint inhibitors targeting PD-L1/PD-1 quickly became of interest to predict therapy response and survival [235]. HPV-related malignancies are associated with a high density of TILs. TILs (CD3+) can be differentiated by specific markers that define corresponding subpopulations such as helper (CD4+), cytotoxic (CD8+) and regulatory T-cells (FoxP3+, Tregs). In a cohort of 120 patients with locally advanced cervical cancer treated with CRT, Enwere et al. were among the first to assess the prognostic impacts of intratumoral CD8+ lymphocyte density and PD-L1 expression. PD-L1 expression (≥1%) was recorded in 88% of patients. Neither the presence of CD8+ lymphocytes nor PD-L1 expression was associated with survival endpoints after primary CRT [236]. Conversely, immunohistochemical detection of CD3 (pan T-cell marker), CD4, CD8, CD20 (B-cells) CD206 (macrophages) and Tregs in a smaller cohort of cervical cancer patients treated with CRT revealed an association of CD8+ TILS with pelvic lymph node metastases, while elevated levels of CD3+, CD4+, CD8+, CD206+ and FoxP3+ lymphocytes were associated with improved PFS and OS [237]. Using flow cytometry, Fan and colleagues investigated the prognostic impacts of PD-1 expression and density of CD4+ and CD8+ lymphocytes in 47 cervical cancer patients. Here, a high PD-1 expression on CD8+ lymphocytes was significantly correlated with a higher incidence of relapse. In addition, a low CD8/CD4-ratio was associated with poor OS in univariate and multivariate analyses [238]. These contradictory results indicate the difficulties in estimating the prognostic and predictive value of immune parameters, which underlie circadian fluxes and other factors. In a meta-analysis including 783 patients from seven studies, high PD-L1 expression was associated with impaired OS (hazard ratio 2.52, *p* = 0.031). Notably, in this meta-analysis only one study investigated PD-L1 expression in patients treated with CRT [239]. Data on the impacts of TILs, as assessed in another meta-analysis, revealed a shift from an immunosuppressive towards a highly immunogenic TME, with high levels of cytotoxic and helper T-cells during carcinogenesis [240]. In this context, Tregs were associated with shortened survival, suggesting their immune-inhibitory potential [241]. Another approach was pursued by Someya et al. Based on CD8, FoxP3, HLA-1, PD-L1, and XRCC4 expression, they classified a set of 100 cervical cancers treated with CRT in “inflamed”, “excluded” and “cold” tumors. “Cold” malignancies were defined by a lack of CD8+ lymphocytes in the tumor and stroma and were associated with a significantly larger tumor volume and a worse DFS, while “inflamed” and “excluded” tumors showed no statistical differences. The authors proposed that antitumor immunity is influenced by radiotherapy, which may overcome the immunosuppressive TME and turn “cold” into “hot” tumors [242]. A sufficient antitumor T-cell response requires an intact antigen presentation mediated by antigen-presenting cells (APCs) and the tumor cell itself. HPV infection modulates the innate immune system by changing human leukocyte antigen (HLA) expression and activates natural killer cell (NK cell) inhibitory receptors [243]. In addition, HPV oncoproteins E5, E6 and E7 promote immune suppression and evasion. For instance, E5 inhibits the transport of major histocompatibility complex (MHC) molecules to the cell surface, which present either viral or tumor-associated antigens [244,245]. Moreover, E6 and E7 interact with type I IFN pathways, mitigating antiviral programs of immune response or upregulating PD-L1 expression (E7) to facilitate lymphocyte dysfunction [234,246].

Other immune components in the TME in cervical cancer are less well understood. Tumor-associated macrophages (TAMs) are a major component of the TME with a pronounced ability to adapt their phenotype and function in dependence of different circumstances within the TME [247,248]. Polarization of TAMs in proinflammatory, classically activated (M1) and immuno-suppressive (M2) subtypes determines their function. The role of macrophages in tumorigenesis and treatment response remains controversial and Janus-faced properties are considered a target for tumor-promoting and antitumor activities [249]. In cervical cancer, high counts of CD68+ (M1) and CD163+ (M2) macrophages are significantly correlated to high-risk HPV infection and carcinogenesis. In transition from CIN to cervical cancer, the density of CD163+ macrophages increases, indicating a polarization towards a M2 subtype. High CD163+ counts were associated with higher FIGO stages and the development of lymph node metastases [250]. Conducting RNA sequencing in 90 cervical cancer patients and a comparative analysis with TCGA data from 286 patients, Qiu et al. investigated the landscape of infiltrating immune cells. In addition to ethnical disparities, they indicated a consistent immune infiltration with high numbers of activated dendritic cells and macrophages in HPV+ tumors. Interestingly, patients with higher Tregs levels tend to have impaired survival, similar to M2 and resting macrophages, which are associated with a poor prognosis [251]. Finally, a low M1/M2 ratio was an independent predictor of poor response and worse survival in a cohort of 84 Italian woman with locally advanced cervical cancer treated with CRT and surgery [252].

Less is known about cancer-associated fibroblasts (CAFs) in cervical cancer. In general, CAFs are associated with malignant progression and an impaired outcome in many different entities [253]. CAFs are known to secrete cyclooxygenase 2 (COX-2) and induce cancer stem cell (CSC)-like activity, promoting apoptotic resistance, proliferation, angiogenesis, inflammation, invasion and metastasis behavior of cancer cells [128]. CAFs derived from five cervical cancer patients were evaluated in vitro for effects in co-culture with Hela cells. Overexpression of transforming growth factor β1 (TGF-β1) and stromal cell-derived factor 1 (SDF-1) was associated with increased cell proliferation, migration, invasion, colony formation and cell cycle progression, while apoptosis was decreased in Hela cells [254]. Furthermore, stromal remodeling processes mediated by CAFs are associated with higher laminin-1 secretion, promoting invasion of CSCC7 cells in vitro [255]. In line with this, Wei et al. very recently identified a pro-metastatic periostin+ subset of CAFs that is correlated with impaired survival. These CAFs activate an integrin–FAK/Src-VE–cadherin signaling pathway to foster metastatic spread [256]. Regarding radiation response, Chu et al. revealed a protective crosstalk between CAFs and cervical cancer cells mediated by different growth factors and radiation response genes, including PDGF, VEGF, EGF, GADD45 and BTG2 [257].

## 8. Microbiota in Cervical Cancer

The increasing knowledge of specific alterations in the composition of the intestinal and cervicovaginal microbiota and its metabolites as well as interactions with neoplastic, epithelial and immune cells in the TME demonstrates the critical role of the microbiome in tumorigenesis and therapy [258]. Notably, global disparities in cervical cancer incidence revealed an association between cervicovaginal infections and the incidence of CINs and cervical cancer [259]. A recent systematic review and network meta-analysis suggested a greater diversity of the cervicovaginal microbiota in HPV+ women, while a greater diversity of cervicovaginal microbiota dominated by non-*Lactobacilli* species expect of *Lactobacillus iners* was associated with high-risk HPV, CINs and cervical cancer [260,261]. The rising alpha diversity is frequently associated with bacterial vaginosis, including *Gardnerella*, *Prevotella*, *Atopobium* and *Sneathia,* as well as the progression from CINs to cervical cancer [262]. *Gardnerella* and *Atopobium* could be involved in forming a biofilm, facilitating viral persistence [263,264]. In contrast, *Lactobacillus crispatus* was associated with HPV clearance [260,261]. Moreover, specific bacterial species such as *Fusobacterium* are under suspicion of promoting the malignant transformation in cervical cancer and other entities [265,266,267].

During CRT, a significant decrease in the bacterial load of cervical cytobrush samples was observed, while neither alpha- nor beta-diversity significantly changed [268]. Sims et al. succeeded in showing that the diversity of the gut microbiota could act as predictor of the survival of cervical cancer patients after CRT. In fecal specimens of 55 patients, the group observed a significant enrichment of *Escherichia*, *Shigella* and *Enterobacteriaceae*. *Enterobacteriales* species were dominant in long-term survivors. In addition, a higher diversity of the gut microbiome correlated with an increased tumor infiltration of activated CD4+ lymphocytes in tumor brush samples, indicating an interaction of the microbiome and the TME [269]. Although longitudinal studies are still lacking and are need to draw a firm conclusion, microbiota appear to contribute to persistent HPV infection and progression of invasive forms of cervical cancer.

## 9. Conclusions

To date, most approaches used to identify and validate robust and feasible biomarkers to predict treatment response and survival for women with cervical cancer in the clinical routine have failed. Regardless, clinical parameters form the basis of treatment decisions and may be used to estimate prognosis. Therefore, HPV infection is an important determinant of response to CRT, impacting a variety of pathways and the immunogenicity of the tumor.

Despite the multitude of dysregulated pathways and extensive research over the last decades, protein biomarkers are not routinely used in predicting therapy response and survival. In contrast, miRNAs and circulating tumor cells or tumor DNA could be used as potential dynamic biomarkers to detect persistent or recurrent disease response in the future. Similarly, this information may help in monitoring and predicting treatment response and outcome, as well as in identifying potential molecular targets.

Immunotherapies have rapidly transformed the treatment of solid tumors over the past decade. In this context, the TME has increasing importance. Tumor development, progression and metastatic behavior are critically influenced by the composition of the TME, resulting in a close association with outcome and survival. Nevertheless, immunotherapy only effects a restricted number of patients and conclusive results of ongoing studies on the implementation of immunotherapy in the primary setting in cervical cancer are still pending.

In summary, accurate prediction of treatment response and survival will help to implement personalized therapies that may improve the treatment of cervical cancer patients. As stated before, the identification of simple, valid and reproducible biomarkers will cover a critical role in this process. At present, however, usage of these markers is neither concise nor clear. Thus, further efforts are needed to implement new approaches and to achieve and expand a holistic view of the multifaceted nature of uterine cervical cancer. Finally, we have to make these achievements accessible to the majority of patients.

## Figures and Tables

**Figure 1 cancers-13-05748-f001:**
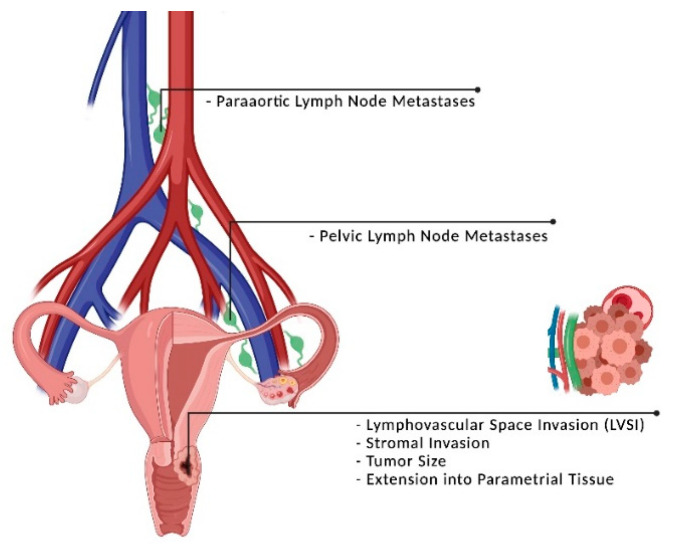
Clinical risk factors in cervical cancer. After primary, curatively intended surgery, Sedlis’ and Peters’ criteria were established to estimate the risk of pelvic and extra-pelvic recurrence. These risk factors include tumor size, deep stromal invasion, involved parametria, lymphovascular space invasion (LVSI) and the presence of lymph node metastases. Created with BioRender.com (accessed on 5 September 2021).

**Figure 2 cancers-13-05748-f002:**
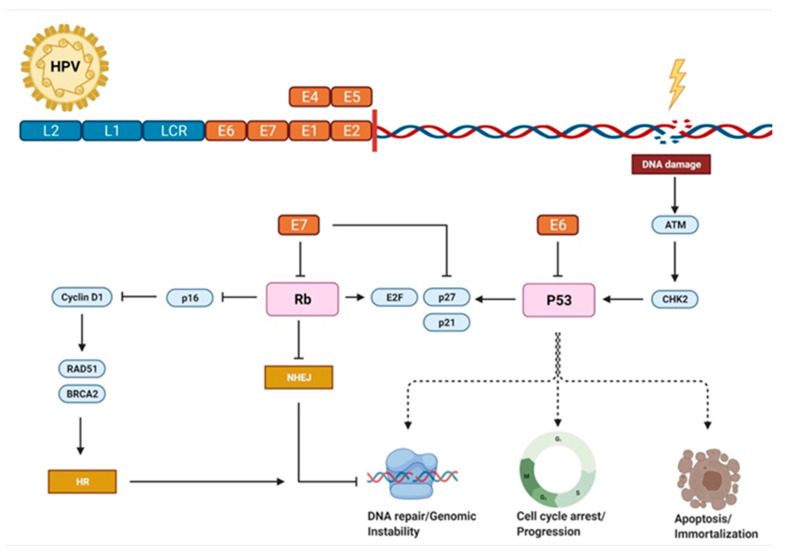
E6 and E7 increase radiosensitivity of HPV-positive cervical cancer cells. An increased sensitivity to radiation by HPV-positive malignancies has been linked to several signaling pathways, including regulation of cell cycle progression, apoptosis and chromosomal instability. Shown is a simplified illustration of DNA damage repair signaling pathways impaired by altered TP53 and p16 activity. After the downregulation of retinoblastoma protein (Rb) by viral oncoprotein E7, increasing p16 levels hamper homologous recombination (HR) regulated via cyclin D1 and subsequently the RAD51/BRCA2 axis. Rb loss further diminishes the non-homologous end-joining (NHEJ) repair pathway. Cell cycle progression and G1-S transition are promoted by the release of the transcription factor E2F and the inhibition of cyclin-dependent kinase (CDK) inhibitors p21 and p27, while in contrast frequently triggered G2/M arrest reorders and arrests cancer cells at vulnerable stages of the cell cycle. Abbreviations: ATM: ataxia telangiectasia mutated; CHK2: checkpoint kinase 2; HPV: human papilloma virus. Adapted from “P53 Regulation and Signaling”, by BioRender.com (accessed on 2021). Retrieved from https://app.biorender.com/biorender-templates (accessed on October 2021).

**Figure 3 cancers-13-05748-f003:**
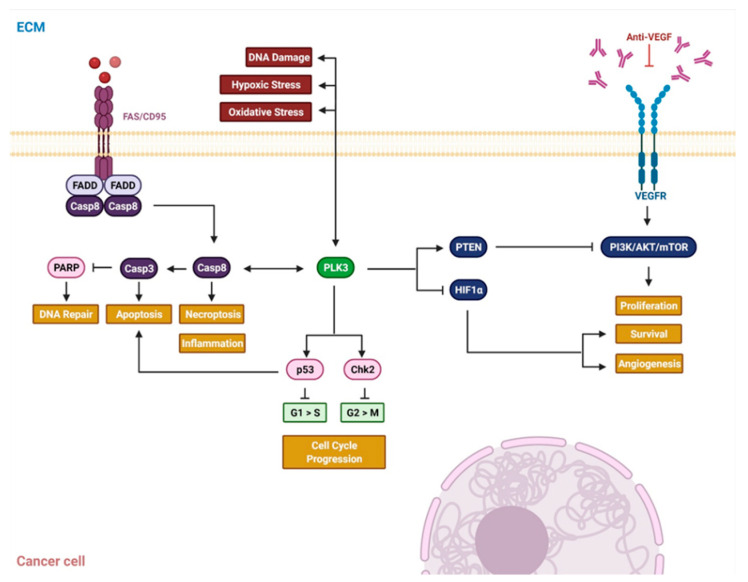
Exemplary illustration of dysregulated signaling pathways and their interactions in cervical cancer. The schematic shows the nexus of dysregulated, activated or discontinued signaling pathways related to PLK3. Upon activation by DNA damage, hypoxic or oxidative stress PLK3 is activated and impacts on apoptosis, cell cycle progression, proliferation and survival. Details are given in the text. Abbreviations: Casp8: caspase 8, Casp3: caspase 3; Chk2: checkpoint kinase 2 FADD: Fas-associated protein with death domain; HIF1a: hypoxia-inducible factor 1-alpha; PARP: poly (ADP-ribose) polymerase; PLK3: polo-like kinase 3, PTEN: phosphatase and tensin homolog; VEGF: vascular endothelial growth factor. Adapted from “FAS Activation Pathway Initiates Cancer Cell Apoptosis” and “Novel Pharmacotherapies for Diabetic Macular Edema (DME)”, by BioRender.com (accessed on 2021). Retrieved from https://app.biorender.com/biorender-templates (accessed on October 2021).

**Figure 4 cancers-13-05748-f004:**
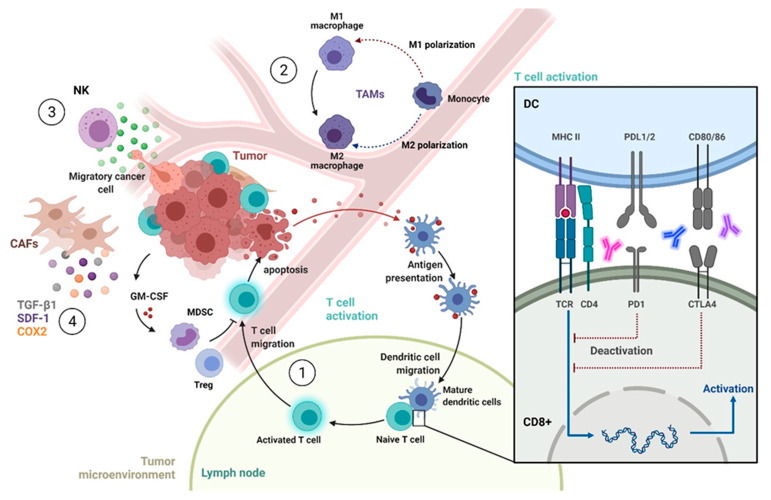
Tumor microenvironment in cervical cancer (1). Immune cells, endothelial cells, fibroblasts, signaling proteins and extracellular matrix molecules within the tumor TME and in lymph nodes impact on tumor biology, therapy response and survival. The adaptive antitumor immune response is mainly mediated by activated CD8+ T cells, while regulatory T cells (Tregs) and myeloid-derived suppressor cells suppress the immune response (2). Further, the polarization status of tumor associated macrophages (TAMs) impact on treatment response and survival, while natural killer (NK) cell activity is modulated and hindered by HPV+ tumor cells (3). Moreover, cancer-associated fibroblasts (CAFs) promote apoptotic resistance, proliferation, angiogenesis, inflammation, invasion and metastatic spread of cancer cells in a paracrine manner (4). Details are given in the text. Abbreviations: CTLA4: cytotoxic T-lymphocyte-associated protein 4; COX2: cyclooxygenase 2; GM-CSF: granulocyte macrophage colony-stimulating factor; PD1: programmed death 1; SDF-1: stromal cell-derived factor 1; TCR: T-cell receptor; TGF-ß1 transforming growth factor beta 1. Adapted from “T Cell Deactivation vs. Activation”, “Tumor-Specific T Cell Induction and Function” and “Cancer Cell Mutations Affect Tumor Microenvironment”, by BioRender.com (accessed on 2021). Retrieved from https://app.biorender.com/biorender-templates (accessed on October 2021).

**Table 1 cancers-13-05748-t001:** Clinical features and postoperative surgical and pathologic risk factors to assess the risk of recurrence in cervical cancer.

Intermediate Risk (Sedlis’ Criteria)	High Risk (Peters’ Criteria)
LVSI plus deep stromal invasion (outer third)	Positive surgical margins
LVSI plus middle stromal invasion (one-third) and tumor size ≥ 2 cm	Detection of pathologically-confirmed lymph node metastases
LVSI plus superficial stromal invasion (inner third) and tumor size ≥ 5 cm	Extension into the parametrial tissue
No LVSI but deep or middle stromal invasion and tumor size ≥ 4 cm

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
