# Peer review of "Molecular Markers to Predict Prognosis and Treatment Response in Uterine Cervical Cancer"

_cancers, 2021, doi:10.3390/cancers13225748_

Round 1

Reviewer 1 Report

In their work, the authors presented a list of molecular markers that should be taken into account in predicting the survival rate of patients with cervical cancer and in assessing the response to therapy. The review is very up-to-date, the authors have comprehensively presented the topic, thanks to which the article will be valuable for the readers of the journal.

Only minor comments:
1. In the title of section 3.4. mention that E6 and E7 are oncoproteins.
2. Explain all abbreviations when they appear for the first time (eg PIK3CA)
3. References should be before the ending period of the sentence - at the moment it is mixed up.

Author Response

Dear Reviewer,

Thank you for the opportunity to re-submit our revised manuscript. In addition, we would like to thank you for your time and effort you put into your kind review! 

  1. We have adjusted the caption.
  2. Thank you for pointing that out. We have double checked all the abbreviations and hope to explain each one correctly. 
  3. We have fixed the errors. 

We look forward to hearing from you and respond to any further questions, comments and concerns you may have. Once again thank you very for your time and effort! 

Yours sincerely 

Maximilian Fleischmann

Reviewer 2 Report

The authors perform a comprehensive review of potential biomarkers and early detection methods that are undergo investigation for cervical cancer.

This is a very well written manuscript that does an EXCELLENT job of summarizing existing data. Useful figures add to readability. Only minor suggestions noted below:

Line 71- please clarify that pembrolizumab is only approved when PDL-1 expression is seen on the tumor (CPS score >1)

Add a sentence about the recent FDA approval of tiostumab into the introduction 

Line 86- change "low risk factors" to "low-risk features"

Line 102- please refer to the more recent publication updating Sedlis's original publication that showed no long-term overall survival benefit 

Line 186- define "low viral load"

Author Response

Dear Reviewer,

Thank you for the opportunity to re-submit the revised manuscript. We would like to thank you again for the time and effort you put into your valuable and kind review. We have been able to incorporate your suggestions in your best interest and to the extent of our knowledge.

  1. Correct! We have added this information.
  2. Excellent suggestion! We have added a short sentence in the introduction to refer to the FDA approval following the efficacy and safety demonstrated by Coleman et al (NCT03438396). Thank you for pointing out this interesting approach!
  3. We have taken this inaccuracy into account.
  4.  We highly appreciate your feedback and agree completely. Interestingly no benefit concerning the OS was demonstrated. We have clarified our statement and included the corresponding publication. 
  5. Here you have raised an interesting point. HPV copy number was estimated by the ratio of the number of HPV reads over the control human gene KLK3 and a cut-off (<4) was defined. We realize that this statement is unfortunately somewhat fuzzy, but not enough data exists to validate it further. 

Once again thank you very for your time and effort as well as your profound and objective review.

Yours sincerely

Maximilian Fleischmann